# Challenges with Methods for Detecting and Studying the Transcription Factor Nuclear Factor Kappa B (NF-κB) in the Central Nervous System

**DOI:** 10.3390/cells10061335

**Published:** 2021-05-28

**Authors:** Marina Mostafizar, Claudia Cortes-Pérez, Wanda Snow, Jelena Djordjevic, Aida Adlimoghaddam, Benedict C. Albensi

**Affiliations:** 1Division of Neurodegenerative Disorders, St. Boniface Hospital Research, Winnipeg, MB R2H 2A6, Canada; mMostafizar@sbrc.ca (M.M.); cperez@sbrc.ca (C.C.-P.); wsnow@sbrc.ca (W.S.); jdordevic@sbrc.ca (J.D.); aadlimoghaddam@sbrc.ca (A.A.); 2Department of Pharmacology and Therapeutics, Max Rady College of Medicine, University of Manitoba, Winnipeg, MB R2H 2A6, Canada

**Keywords:** nuclear factor kappa B, transcription factor, neurological disorders, in vitro and in vivo methods

## Abstract

The transcription factor nuclear factor kappa B (NF-κB) is highly expressed in almost all types of cells. NF-κB is involved in many complex biological processes, in particular in immunity. The activation of the NF-κB signaling pathways is also associated with cancer, diabetes, neurological disorders and even memory. Hence, NF-κB is a central factor for understanding not only fundamental biological presence but also pathogenesis, and has been the subject of intense study in these contexts. Under healthy physiological conditions, the NF-κB pathway promotes synapse growth and synaptic plasticity in neurons, while in glia, NF-κB signaling can promote pro-inflammatory responses to injury. In addition, NF-κB promotes the maintenance and maturation of B cells regulating gene expression in a majority of diverse signaling pathways. Given this, the protein plays a predominant role in activating the mammalian immune system, where NF-κB-regulated gene expression targets processes of inflammation and host defense. Thus, an understanding of the methodological issues around its detection for localization, quantification, and mechanistic insights should have a broad interest across the molecular neuroscience community. In this review, we summarize the available methods for the proper detection and analysis of NF-κB among various brain tissues, cell types, and subcellular compartments, using both qualitative and quantitative methods. We also summarize the flexibility and performance of these experimental methods for the detection of the protein, accurate quantification in different samples, and the experimental challenges in this regard, as well as suggestions to overcome common challenges.

## 1. Introduction

In 1986, Ranjan Sen and David Baltimore [1] identified a transcription factor [2] by using a simple gel electrophoresis mobility shift assay that showed binding to the sequence GGGACTTTCC, the enhancer element of the immunoglobulin kappa light chain of B cells [1,3,4]. Subsequent work has shown that this transcription factor and its associated proteins are expressed in all cell types [2,5]. Baltimore and colleagues named this protein nuclear factor kappa B (NF-κB). NF-κB is made up of the Rel family [5] of DNA binding proteins [6]. There are five different forms of Rel proteins, often divided into two classes [2,6]. The first class comprises Rel A (p65), c-Rel, and Rel-B proteins that do not require proteolytic processing [2,3,5,6,7,8]. The second class includes NF-κB1 (p105) and NF-κB2 (p100), which are found as large precursor proteins [9,10,11]. These precursor proteins undergo proteolytic processing to form the p50 and p52 subunits of NF-κB, respectively [2,5,6,10].

The subunits of NF-κB can be found in homodimer or heterodimer forms, which are structurally similar to the Rel homology domain (RHD) [12,13]. This structural resemblance assists in dimerization and DNA binding [14,15]. In the inactive state, NF-κB dimers are associated with the inhibitory κB protein (IκB) [7,10], which masks the nuclear localization signal. However, once triggered by a series of upstream molecules from inside or outside the cell, IκB interacts with IκB kinase (IKK) [3], resulting in the phosphorylation, ubiquitination, and then degradation of IκB [16]. These reactions cause the NF-κB dimers to dissociate from IκB and translocate to the nucleus [17,18,19], where they bind to specific DNA sequences and promote the transcription of target genes [8] (see Figure 1). Overall, NF-κB is responsible for controlling the expression of more than 400 genes that are involved in immune and inflammatory responses, cell proliferation and apoptosis, as well as stress responses [20,21]. Among the neuronal gene targets of NF-κB in the CNS, and those relevant to the Alzheimer disease (AD) brain, are the following: brain-derived neurotrophic factor (BDNF), copper-zinc superoxide dismutase (CuZnSOD), manganese superoxide dismutase (MnSOD), calcium-calmodulin kinase II (CaMKII), postsynaptic density protein-95 (PSD-95), early growth response (Egr) factors, and cyclo-oxygenase-2 (Cox-2) [22]. Although not a confirmed gene target of NF-κB, the transcription factor cyclic adenosine monophosphate (cAMP) response element-binding protein (CREB) is also an important regulator of learning and memory and is responsive to changes in NF-κB activity through, in part, shared associations with the co-activator CREB-binding protein (CBP) [22,23].

There are the following two well-known signaling pathways for NF-κB activation: the canonical pathway and an alternative or non-canonical pathway [6,7]. The canonical pathway is stimulated by ligand-receptor binding, where receptors can be antigen, cytokine, or pattern-recognition receptors, etc. [24,25]. This biological cascade is key for the initiation of innate immunity and inflammation [2,15] and it is NF-κB p50 subunit-dependent. On the other hand, the non-canonical pathway is highly dependent on the IKKα dimer [25]. This pathway is crucial for the development of lymphoid organs [9] and it is NF-κB p52 subunit-dependent.

Furthermore, NF-κB is induced by different stressors, such as environmental, mechanical, chemical, and microbiological, etc. [5,6]. Specifically, in the central nervous system (CNS), some stressors that trigger NF-κB activity are β-amyloid [26], opioids [27], sleep deprivation [28], and synaptic activity [29]. NF-κB is in fact characterized as a family of inducible transcription factors [6] that are involved in multiple cellular activities [19] such as DNA transcription, cytokine production, cell survival [6], innate and adaptive immune responses, and organ development [2]. Evidence also shows a dual nature for NF-κB in the nervous system [9]. For example, the activation of NF-κB in neurons leads to cell survival, whereas glial activation initiates pathological inflammatory pathways [19,30].

Within the CNS, NF-κB is well known for its inflammatory and immune functions [31]; however, NF-κB activity has also been implicated in normal brain functions such as learning and memory [32,33], neurogenesis [34], and neuritogenesis [35]. Studies have shown that inhibiting NF-κB activation modulates synaptic plasticity [29,32,33]. Due to its constitutive expression in neurons, this protein is abundant in brain tissue, where the most abundant subunits found are NF-κB p50 and p65 [36,37]. In addition to its localization in neurons, NF-κB also resides in non-neuronal cells, including astrocytes and microglia [38,39]. Together, these cell types are involved in the development and coordination of the cellular response to injury, and the pathology of the CNS [31].

NF-κB-mediated brain inflammation has been linked with both aging and neurodegenerative disorders [40,41], including Alzheimer’s disease (AD) [42], which is a progressive brain disorder that is most prevalent in individuals between 65 and 85 years old [43]. This disorder is often associated with memory loss, language problems, and other diminished cognitive abilities [44]. In the brains of those with AD, the levels of NF-κB p65 are preferentially elevated within the neurons and astrocytes near the amyloid β plaques, a classic neuropathological hallmark of AD. Moreover, NF-κB activation is induced in cultured neurons when exposed to amyloid β-peptide or secreted amyloid precursor protein (sAPP), further supporting a role for NF-κB in AD-related neuropathology [26]. There is also an increase in the levels of NF-κB in the cholinergic neurons in the basal forebrain of AD patients [45].

NF-κB activation is stimulated by genotoxic stressors such as ionizing radiation, reactive oxygen species (ROS), DNA double-strand breaks (DSBs), and replication stress [46,47]. This involves an array of signaling cascades from the nucleus to the cytoplasm and direct nuclear NF-κB regulation. NF-κB is activated by DNA damaging drugs such as temozolomide-induced SN1-methylation, cisplatin-induced DNA cross-linking, and IR-induced DSB [46,47]. IR-induced DSB stimulates ATM, which is a crucial kinase associated with NF-κB activation following DNA damage. This was first observed in 1998 when the activation of NF-κB by IR was seen to be reduced in human cells with ATM deficiency (A-T cells) [46]. NF-κB is known to be involved in the transactivation of genes that participate in DNA damage response (DDR) pathways. DDR is crucial for cellular homeostasis, genomic stability and survival, otherwise it can lead to the development of cancer and aging [46,48].

In terms of DNA DSBs, NF-κB activation involves the following two key regulators: ataxia and telangiectasia mutated (ATM) and nuclear NEMO. Upon DSBs, NEMO interacts with the following proteins to activate IkK for the nuclear transport of NF-κB: leucine-L, lysine-K, protein rich in glutamate-E, serine-S (ELKS), ATM, and TAK1. The C-terminal zinc finger (ZF) domain of NEMO is required for the activation of NF-κB when DBS is induced by DNA-damaging agents [46,47,48,49]. Downstream of the signaling pathway regulation is conducted by ATM- and Rad3-related protein (ATR) in a more complex and protein-specific manner. ATR mediates its process through check-point kinases and it induces the Chk1-mediated phosphorylation of p65 (Thr505) and p50 (Ser329), promoting cell death. This pathway is important for the therapy of glioblastoma multiforme (GBM). As mentioned above, NF-κB is activated by temozolomide (TMZ), which is the first-line therapy for GBM. This drug induces reactive oxygen species (RS), which activates ATR through cytotoxic O6-methylguanine (O6-meG) lesions [47,50].

The activation of NF-κB in these multiple signaling pathways is complex and there are various crosstalk connections. The activation of NF-κB has been seen in lymphoma, leukemia cell lines, breast, and lungs. Poor outcomes with increased activation of NF-κB has been demonstrated with ovarian cancer and glioblastoma, and the inhibition of this activity has shown an anti-tumor response. These findings indicate more research is needed to understand the signaling pathways in detail for the development of cell-specific inhibitors [47,50].

Collectively, these observations signify the central involvement of NF-κB in complex signaling processes under both normal and pathological conditions; given its broad impact physiologically, evidence that shows that NF-κB dysregulation (across potentially multicellular pathways) is associated with a multitude of diseases is, therefore, not surprising [19,51]. This protein is constitutively activated in neurodegenerative diseases, autoimmune, cancer, and inflammatory conditions, etc., and its ability to control multiple genes in human disease makes it a novel therapeutic target [13,44]. Also, studies have shown that NF-κB activation [6] is normally tightly regulated [5,14], consistent with it being considered a “master regulator” [52].

Thus, given the multiplicity of cell types, gene targets, biological processes, and pathological conditions in which the NF-κB signaling pathway has been implicated in the CNS, there is considerable interest in the methods available to investigate this pathway as well as to understand the potential limitations of the currently available methods. This article will discuss the various qualitative and quantitative methodologies for NF-κB detection (e.g., localization, gene expression, protein activation) in the CNS tissue (e.g., cytosolic, nuclear, mitochondria, and total lysate). This paper summarizes and compares the different detection assays, with a focus on applying them in the context of neuroscience research. In addition, the challenges encountered in different tissue types within the CNS are highlighted, as well as suggestions for overcoming these challenges.

## 2. Identification of NF-κB in the CNS

Due to its presence in the synapse and suspected roles in multiple neurophysiological functions, [53], NF-κB has been a topic of increasing interest in neuroscience research. In the CNS, NF-κB is present in various DNA binding forms, among which the major subunits present are p50 and p65 [36,53,54]. High basal NF-κB activity is found in the glutamatergic neurons of the hippocampus (granule cells and pyramidal cells of CA1,CA3) and cerebral cortex (layers 2, 4 and 5), as detected by immunohistochemistry, as well as Western blotting and gel shift assays [36]. In addition to this high level of basal activity, the electro mobility shift assay (EMSA) has also shown that this transcriptional regulator is abundant in brain tissues in an inducible form [36]. Individual NF-κB subunits and their post-translational modifications have been detected predominantly by Western blotting [55,56]. NF-κB localization and translocation to the nucleus are also studied using immunostaining techniques, such as immunohistochemistry and immunocytochemistry for detection in tissues and cells, respectively [37]. The quantification of the nuclear translocation of NF-κB has been established by using high-content fluorescence microscopy and imaging flow cytometry [57]. On the other hand, EMSA [37] and commercial DNA binding ELISA (such as TransAM^®^ NF-κB activation assays from Active Motif^®^) have been widely used to assay NF-κB activation [58,59,60,61].

The integration of different experimental methods to study NF-κB transcription activation and expression both in vivo and in vitro, quantitatively and qualitatively, can help in understanding the molecular mechanisms involved in various activities. For instance, the use of enzymatic assays, such as an in vitro assay for IKK kinase activity [62,63], can be used to complement NF-κB activation results. There are of course several considerations for selecting any experimental methods to detect NF-κB and its signaling. Moreover, each aspect of the detection process should be well thought out to ensure accurate interpretation of the data. For example, NF-κB subunits are not exclusively localized in the nucleus of a cell [64], and NF-κB signaling involves several steps, as follow: the activation of IKK [65], IκB degradation [66] and ubiquitination [67], post-translational modification of the NF-κB subunits [68,69], and translocation of the NF-κB dimers to the nucleus from the cytoplasm and their binding to κB-sites [70]. All these steps need to be considered as one attempt to detect NF-κB, and measure its action and/or abundance. Every method has its limitations, advantages and disadvantages; weighing the pros and cons of any method used for NF-κB detection in the CNS will lead to a more accurate interpretation of the results. Figure 1 gives a clear representation of the different NF-κB subunits that are being targeted for the detection of the protein through different molecular assays.

### 2.1. Gel Electrophoresis Mobility Shift Assay (EMSA) for NF-κB

The gel electrophoresis mobility shift assay (EMSA) is designed to analyze the interaction of DNA binding proteins with DNA sequences [71,72]. EMSA has been widely used as a sensitive index of nuclear NF-κB activity [73,74] due to its simplicity, sensitivity, and robustness measures (Table 1) [61,73]. In particular, this technique is used as a quantitative index [61] to detect the binding of active NF-κB to its DNA recognition sequence [74,75].

The crucial part in this technique is the purification of the nuclear and cytosolic (latent form of NF-κB) extracts that consist of DNA binding proteins [75]. This extract is then subjected to electrophoresis using optimal conditions [61,73], and polyacrylamide or agarose gels. The NF-κB:DNA complexes can be identified by autoradiography—in particular ^32^P-labeled nucleic acids [37], by chemiluminescent systems of biotin or digoxin conjugated to DNA [76,77], or by the direct detection of DNA fluorescent dye conjugates by scanning the gel [78]. In addition, a titration protocol is utilized to assess the protein concentration required to interact with a constant amount of nucleic acid. This step is crucial and, if not optimized properly, can present as a difficult challenge. For example, protein concentrations of 5 µg/µL to 8 µg/µL in the brain tissues (cortical, hypothalamus) work well [20]. A slight change in the protein concentration outside this range can make a tremendous difference in the signal detection. For some brain compartments or tissues/cells (neurons, astrocytes), more starting material is required to obtain an optimal range of protein concentration, such as the hypothalamus compared to the cortex. Given the sensitivity of this technique to protein concentrations, different titration protocols are required, not only based on the type of biological material (e.g., tissues versus cells, such as primary neuron cultures), but also on the specific brain region (e.g., cortex, hippocampus, cerebellum and hypothalamus) [20]. The components for trituration are premixed in individual tubes at constant concentrations and volumes to minimize sample variation. The protein is added after the non-protein components are mixed and reach the reaction temperature. This is done so that the protein meets all the components at their final concentration. Among all these, it is important to reach an equilibrium state, otherwise it will lead to irreproducible results. The timing for equilibration depends on the reaction conditions (e.g., protein concentration, nucleic acid concentration, temperature, salt concentration) and identities of the reacting molecules. This can be resolved by aliquoting each sample into two tubes and subjecting them to different equilibration times, and then assessing the mixture under identical conditions [79].

EMSA has limitations as well, such as this technique does not provide information regarding the location of the nucleic acid sequences to which the protein is associated (Table 1) [79]. It does not provide information on the molecular weight and identities of the protein in a protein-nucleic acid complex (Table 1) [44,79]. Also, the samples are not in an equilibrium state in electrophoresis, hence rapid dissociation prevents detection, and slow dissociation will undermine the binding density [75]. On the other hand, there are some species that are stable in gels and not in solution, so a short electrophoresis time will allow distribution of the species in the sample at the start of electrophoresis [79].

### 2.2. Oligonucleotide-Based Chemiluminescent Enzyme-Linked Immunosorbent Assay (ELISA)

The TransAM^®^ NF-κB activation assay (Active Motif^®^) and NF-κB p50/p65 EZ-TFA transcription factor assay (MilliporeSigma) are commercial chemiluminescent/colorimetric ELISA kits that allow quick and easy measurements of the activity of the NF-κB family members p50, p52, p65, c-Rel, and Rel B. These assays are non-radioactive, sensitive ELISA-based methods for detecting specific transcription factor DNA binding activity [80,81] in nuclear extracts.

In this assay system, NF-κB subunits are identified [21] by a double-stranded oligonucleotide probe containing the consensus binding sequence for NF-κB (5′-GGGACTTTCC-3′) [82] using multi-well plates. The oligonucleotide probes are then incubated with the nuclear extract [81,83]. The active NF-κB forms in the nuclear extract will bind to the capture probe, and this binding will be detected using a primary anti-NF-κB antibody, followed by a secondary antibody conjugated to horseradish peroxidase. Finally, the results are quantified by a chromogenic or chemiluminescent reaction [81,82]. In other words, this assay works as an EMSA assay in a 96-well plate.

This DNA binding method for site-specific NF-κB detection is very sensitive, efficient, and quantitative (Table 1) [83]. This technique has been employed successfully in experimental research to assess the activity of transcriptional proteins in the hippocampus via phosphorylated NF-κB p65, transcriptional proteins in mouse cortical neurons by examining the NF-κB p65 subunit [84,85], and in mouse microglia cell lines [86].

### 2.3. Luciferase Reporter Assay

Reporter gene assays use specific regulatory sequences attached to the genes of an organism that can be quantified through fluorescence emission. Similarly, distinguished luciferase and fluorescent protein-based techniques have been employed for monitoring the signaling pathways in aging [87]. The luciferase assay is an effective method to monitor gene expression through bioluminescence (Table 1) [88]. The luciferase reporter assay is a traditional method used for assessing signaling pathways to measure the levels of activated NF-κB in in vitro models [87,89]. Luciferase is an enzyme produced by the firefly family Lampyridae [90]. Luciferases have been widely used in this assay due to their capability for highly sensitive quantification [91]. These enzymes catalyze the conversion of luciferin to an excited electronic state, resulting in luminescence when luciferin comes back to its ground state [92]. The luminescence is then detected by a luminometer. This assay has proven to be extremely useful in observing the effects of drugs on transcriptional levels [93,94,95]. The luciferase assay has been mostly used in monitoring NF-κB activation in cultured cells, such as primary mouse astrocytes and microglia [89,92]. Usually, a non-viral system is employed that encodes the firefly luciferase reporter gene and this is then used for monitoring NF-κB activation. This gene is kept under the control of a minimal cytomegalovirus (CMV) promoter with tandem repeats of the NF-κB transcriptional response element. Upon a trigger, NF-κB signaling is activated and causes the reporter gene to be transcribed [87]. This leads to the nuclear translocation of NF-κB transcription factors. In-vivo visualization is monitored by bioluminescence emission by the tissues upon luciferin administration [87,96].

### 2.4. Immunostaining

Immunostaining (immunofluorescence, immunohistochemistry, immunocytochemistry) has been an important method for neuroscientists over the years, where it allows one to characterize the distribution and performs semi-quantification of a specific protein and/or molecule in tissues and cultured cells. This type of visualization can supply a lot of anatomical detail through antibody immunostaining (Table 1). The use of this technique is in demand for the diagnosis, treatment of disease, and exploration of pathogenesis [97]. Immunohistochemistry has been employed in research to identify proteins with a specific primary antibody, followed by a secondary antibody for visualization using microscopy [98]. This technique is carried out in the following two phases: slide preparation that consists of tissue fixation, and detecting the protein of interest [98,99].

For the localization and potential quantification of NF-κB subunits using this technique, the brain regions are first fixed with 4% paraformaldehyde [98,100], embedded in paraffin, then cut into sections on a microtome or a cryostat, and dehydrated [101]. Antigen recovery may need to be performed in order to expose the epitope and to ensure an adequate signal [101]. The images of brain slices are visualized under the microscope for interpretation and quantification of NF-κB activity (depending on the specific antibody used), with filtering for DAPI, FITC/Alexa Fluor 488, and Texas Red/Alexa [102]. Besides fluorophores, there are several chromogens that are also used for detection, such as DAB, DAB +Ni, AEC, VIP, NOVA RED, etc. [103].

The presence and activation of neuronal NF-κB has also been detected in brain cultures. Cells such as neurons, astrocytes, or microglia were immunostained for NF-κB p65 and then co-stained with antibodies to specific cell types. These brain cells showed different responses to p65 signaling when using different methods. When viewed under the microscope, significant amounts of p65 were present in the cytoplasm and nucleus of these cells in culture. The expression of p65 was determined with Western blotting and immunostaining methods. In brain cell cultures stimulated/supplemented with TNFα, the nuclear translocation of p65 was not appreciable in neurons, but a strong signal was seen upon immunostaining with a neuron-specific βIII-tubulin antibody. The scenario was different for microglia, which have very clear p65 staining in TNFα-stimulated brain cultures [37]. Another study was carried out to investigate the potential involvement of the NF-κB signaling pathway in the presence of nilotinib (selective tyrosine kinase inhibitor) in astroglia from 3xTg (AD model mice). The study reported that the NF-κB signaling pathway is involved in the bioenergetics of 3xTg astroglia and that the drug nilotinib increased NF-κB p50 subunit translocation to the nucleus of the 3xTg astroglia, suggestive of increased activation of NF-κB. These data were generated using immunofluorescence [104] (see Figure 2).

This technique seems to be straightforward, but is quite challenging and can present significant bias, such as reaction bias and interruption bias (Table 1) [98]. Reaction bias is described as pitfalls at the technical level, such as specimen fixation, tissue processing, detection system, and tissue pretreatment. Similarly, pitfalls at the design and interpretation level are described as interpretation bias, which includes the specificity and sensitivity of the antibodies [98].

### 2.5. Western Blotting

Western blotting is used to investigate protein abundance, kinase activity, cellular localization, protein-protein interactions, monitoring of post-translational modifications, events of cleavage, phosphorylation, glycosylation, and methylation [105]. This technique is employed for the identification of the protein of interest in a mixture of molecules with various molecular weights, and by subjecting the physiological material to gel electrophoresis [106]. The proteins are then transferred to a membrane (nitrocellulose, or polyvinylidene difluoride (PVDF)), producing a band for each type of protein [107]. The membrane is then incubated with specific primary antibodies against the protein of interest, followed by secondary antibodies with labels (Alexa Fluor dyes, biotin) [100]. The antibodies bound to the protein of interest are observed under a chemiluminescence image analyzer. The thickness of the band and the density of the signal correlates to the amount of the protein present, which is normalized based on a housekeeping gene or total protein [106,107].

The nuclear localization of NF-κB can be assessed semi-quantitatively by Western blotting through various, but specific, NF-κB molecules that are upstream and/or downstream in the signaling pathway [107,108]. During the initial activation of NF-κB, the IKK complex is activated, releasing IKKα and IKKβ [109]. The canonical pathway is activated through the degradation of IκBα, which causes the translocation of NF-κB subunits to the nucleus, particularly the p50/RelA dimer. On the other hand, the non-canonical pathway is responsible for the activation of the NF-κB subunit RelB/p52 via a p100-inducible process [109,110]. The aforementioned study reporting increased NF-κB translocation to the nucleus in 3xTg verses C57/BL6 with nilotinib also used Western blotting techniques (see Figure 3) [104].

One of the most investigated NF-κB subunits with Western blotting is p65. The phosphorylation of this protein causes a conformational change, resulting in stability, protein interactions, and ubiquitination of the protein [111]. Neurons have constitutive NF-κB activities, which are detected by commercially available p65 antibodies [112]. Studies have shown most of these antibodies to be specific towards the ‘activated’ form of p65. On the other hand, recent work has shown that some commercially available p65 products show complex binding with multiple proteins in Western blot analyses [112]. One of the major limitations of Western blotting is the requirement of a specific antibody to detect the protein of interest; several other factors include antibody efficiency, membrane exposure time, and background exposure, to name a few (Table 1) [113]. Other challenges exist as well. When the protein undergoes post-translational modification, a specific primary antibody is required for the detection of the phosphorylated form. There are some proteins expressed in certain cell lines and tissues that do not have specific primary antibodies, hence it is not possible to detect that protein [113]. For example, there are several antibodies available for p50 and p65 to detect proper localization, and the distribution and activity of NF-κB, but few are specific. A study was conducted to determine the specificity of antibodies towards p50 and p65. Around six antibodies from different companies have been tested for p65 using Western blots and immunocytochemistry (ICC). Cytokine tumor necrosis factor alpha (TNF-α) has been used with embryonic fibroblasts (MEFs) for the activation of NF-κB. This activation was then detected with p65 antibodies via immunocytochemistry. Stimulated MEFs showed signals using the antibodies and non-stimulated ones did not show cytosolic signals, except for the antibodies sc-8008, sc-372 and E498. On the other hand, sc-7151 showed a single band at p65 with cross-reactivity in immunocytochemistry, and E498 did not mark any bands. For the Western blotting, the antibodies showed a proper signal at p65, except sc-372 with no cytosolic immunoreactivity in immunocytochemistry [112].

### 2.6. Polymerase Chain Reaction (PCR)

The polymerase chain reaction (PCR) has proven to be a very useful biological tool [114]. However, when using abbreviated terminology, scientists sometimes get confused. For example, the RT-PCR (reverse transcription PCR), qPCR (quantitative real time PCR) and RT-qPCR (reverse transcription quantitative real-time) terms are not all the same. RT-qPCR is used for the detection and quantification of gene expression. It is usually considered as a multiplex system for identifying multiple genes [115]. This makes RT-PCR a popular technique for the study of the gene expression profile in neurological disorders. RT-PCR requires a relatively low quantity of mRNAs in tissue samples for the detection of gene expression. The identification of a gene at a stable level is difficult to detect because of fluctuations in the mRNA synthesis, which happens due to changes in cell behavior. Thus, reference genes, also known as ‘’housekeeping genes’’, are required for analysis by RT-qPCR for the normalization to the expression level of genes not expected to fluctuate with a given experimental condition [116]. The use of an ideal reference gene for data normalization that will be expressed constitutively in all samples regardless of tissue type, disease state, or experimental conditions is required for accurate quantitative expression as compared to target genes. The selection of an “ideal” reference gene, however, depends on the experimental condition of interest. For example, glyceraldehyde-3-phosphate dehydrogenase (GAPDH) is a widely used reference gene for the normalization of brain samples. Increased aggregation in the nucleus and increased expression of this gene in the tissues of AD patients, however, have been demonstrated, making it inappropriate for normalization purposes in this population. Along with GAPDH, β-actin is also used frequently as a reference gene, but β-actin has also been shown to be affected by experimental conditions and clinical conditions (e.g., asthma). These genes may not be suitable for normalization in various cases, and non-validated genes used as a reference gene or housekeeping gene may lead to inaccurate results. Proper study and validation of the reference is required before its use in any experiment [116].

In one study, 12 reference genes were analyzed from the brain samples of individuals diagnosed with AD, and from control groups. CYC1 and EIF4A2 were found to be the best reference genes for experiment after analysis with geNorm and NormFinder algorithms [117]. Another study reported 15 reference genes for the gene expression profile on human brains relating to neurodegenerative diseases. These reference genes can be used for data normalization in the brain samples from patients with AD, progressive supranuclear palsy (PSP), PD, and multiple system atrophy (MSA). These 15 reference genes include GAPDH, ACTB, ribosomal protein large 13 (RPL13), hypoxanhine phosphoribosyl transferase (HPRT1), cytochrome C1 (CYC1), topoisomerase 1 (TOP1), eukaryotic translation initiation factor 4A2 (EIF4A2), β-2-microglobulin (B2M), pumilio homolog 1 (PUM1), TATA box-binding protein (TBP), ubiquitin C (UBC), cyclophilin A (PPIA), succinate dehydrogenase complex subunit A (SDHA), ATP synthase H+ transporting mitochondrial F1 complex beta polypeptide (ATP5B), and ubiquitin-conjugating enzyme E2D2 (UBE2D2) [116,118].

PCR is carried out through the use of fluorophores followed by an analysis of the melting curves of the amplified gene [119]. This method has been employed for exploring the transcriptional gene regulation activity of NF-κB. The fluorescent detection of qPCR has allowed one to observe the IκB mRNA that is correlated with NF-κB activation in different cellular models [114]. In a recent study, the role of NF-κB in brain injury after aneurysmal subarachnoid hemorrhage (SAH) has been determined by assessing the change in NF-κB DNA binding activity in the brain. Along with this, the effect of pyrrolidine dithiocarbamate (PDTC) in brain injury was also investigated. RT-PCR has been used in this context to determine the levels of TNF-α, IL-1b, and ICAM-1 mRNA expression [120]. PCR is rapid, sensitive, and applicable for primary cells and frozen samples (Table 1) [114,115,119]. However, qPCR has its own limitations, including that the variation in gene expression increases with rises in the cycle number [121]. Certain biological samples have inhibitors that are endogenously expressed; hence, the PCR is subjected to inhibition by these compounds from biological samples. In clinical and forensic applications, qPCR is usually blocked by inhibitors in body fluids, such as urea and hemoglobin. For food applications these inhibitors may be organic and phenolic compounds. To resolve this problem, alternative DNA polymerases are used that are resistant to certain inhibitors, such as Tfl and Pwo [122]. Moreover, qPCR methods also have a very high possibility of false negative or positive results (Table 1) [121]. This is usually resolved by visually cross-checking amplification, melting curves and any calculations based on these curves [122].

### 2.7. Chromatin Immunoprecipitation (ChIP)

Several biological processes and signaling pathways in our body involve protein-DNA interactions such as the regulation of gene expression, transcription, DNA repair, DNA replication and recombination, chromosome segregation and instability, cell cycle progression and epigenetic silencing. Therefore, it is important to understand the interactions that are involved in triggering and driving such biological processes. There are various methods through which we can determine or detect these interactions, including EMSA, DNAse footprinting and chromatin immunoprecipitation (ChIP) assays. EMSA has proven to be a useful method for detecting these interactions, but EMSA comes with limitations such as that samples are not in equilibrium and some complexes are more stable in the gel than in free solution. ChIP assays eliminate these limitations [123]. This technique is popular for its effective and unbiased changes with chromatin in vivo that occur due to extracellular signals during development and differentiation [124]. A study has been conducted to observe the functional contribution of p65 DNA binding capacity and found that the functional contribution of p65 DNA binding and dimerization in p65-deficient human and murine cells with single amino acid mutants prevents DNA binding (p65 E/I) or dimerization (p65 FL/DD) [125]. Therefore, the DNA binding of p65 is required for RelB-dependent stabilization of the NF-κB p100 protein. Also, the anti-apoptotic function of p65 and expression of TNF-α–induced genes are heavily dependent on p65′s ability to bind DNA and to dimerize. The ChIP assay concluded that improper DNA binding and dimerization reduce the chromatin association of p65 [125].

This technique has been used in several mammalian cell lines, whole mouse embryos, as well as yeast. This assay has also been used to measure the long-range enhancer binding of specific proteins and allele-specific transcription factor binding patterns [123,124]. The first step in this assay is to crosslink the DNA to the protein by treatment with formaldehyde, followed by isolation of the DNA-protein complexes and fragmentation of the isolated DNA into 1–2 Kb fragments. The fragmentation is carried out by micrococcal nuclease or sonication. These fragmented DNA are then immunoprecipitated with specific antibodies. These antibodies could be directed towards transcription factors, such as for NF-κB (e.g., p65), modified histones, coactivators, or corepressors. These immunoprecipitated DNA-protein complexes are de-crosslinked followed by PCR with specific primers to the promoters [123,124,126]. There are the following three steps that are needed for the optimization of the ChIP assay: fixation time, amount of antibody, and amount of input material. The known target gene for the DNA binding protein of interest serves as a positive control [124]. There are the following four different types of ChIP assays: X-ChIP, N-ChIP, ChIP cloning and ChIP-CpG microarray. X-ChIP is used for investigating the interactions of proteins bound to DNA, such as histone-chromatin interactions, whereas N-ChIP is highly specific for antigens and is used towards the analysis of histones and its isoforms. On the other hand, ChIP cloning is used for the isolation of target genes, and ChIP-CpG microarray for unknown DNA sequences interacting with a known protein (Table 1) [126]. Although ChIP assay is a useful tool to detect the interactions of DNA-bound proteins, this assay is more of a qualitative approach than quantitative. The data can be misleading, as the reaction with formaldehyde is never complete, variability in cross-linking between the DNA and the target protein will occur among samples, and variability in the immunoprecipitation of antibodies will occur (antibodies may not immunoprecipitate all the antigens from among the samples) (Table 1) [126].

### 2.8. High-Throughput Microscopic Imaging System for NF-κB Detection

NF-κB nuclear localization is determined quantitatively by various molecular approaches as described above (Western blotting and EMSA), but a significant drawback of these techniques is the lack of information for sample heterogeneity [108]. These approaches also have poor sensitivity and procedures can be lengthy [127], which can be overcome by using microscopic analysis. A major advantage of microscopic evaluation is the assessment of heterogeneity within a sample [108]. NF-κB activation has been detected by various high-content, automated imaging systems, such as high-content screening (HCS), high-content analysis (HCA), high-content imaging (HCI), or image cytometry (IC) [128]. These approaches quantitatively measure the translocation of the NF-κB protein from the cytoplasm to the nucleus [127], and are well suited for adherent cell lines and tissue sections rather than cells in suspension [108]. This provides a platform to measure multiple cellular characteristics for NF-κB activity in a non-biased fashion and is statistically more accurate [129].

The activation of NF-κB can be triggered through various stimuli and by different dynamics. To address these variations, it is very important to analyze single-cell NF-κB dynamics in a large cell population. There are several factors that are involved with the translocation of NF-κB to the nucleus—under this scenario, high-throughput screens are necessary in cell lines for the purpose of compound screening and functional genomics [129]. High-throughput assays are based on tagging the nucleus of a cell with a nuclear marker, such as Hoechst, DAPI, or DRAQ5. Nuclear markers make it possible to detect and analyze the translocation of NF-κB [127]. The nucleus of the cell is detected using an image analysis system and the nucleus is specified with a primary mask followed by a secondary mask in the cytoplasm. The labeled NF-κB is quantified within the secondary mask (nucleus/cytoplasm). NF-κB is detected by an image analysis system by the intensity of NF-κB in the nucleus and the cytoplasm [128]. Then, the translocation value is determined by measuring the difference between the average intensity of NF-κB in the cytoplasm and the nucleus (cyto-nuc difference). These methods are helpful in screening a library of compounds associated with NF-κB translocation in various cell types that can be quantified with some modifications [128]. A study has been conducted on high-throughput detection for NF-κB translocation in embryonic fibroblasts (MEFs) from a GFP-p65 knock-in mouse model. They took into account the nuclear-to-total ratio (NT) of the fluorescent signal that determines the total amount of NF-κB per cell and the fraction that relocates in the nucleus over time [130].

On the other hand, flow cytometry is used for the detection of NF-κB phosphorylation events in heterogenous populations by co-staining with primary antibodies and/or antibodies for immunoprecipitation [131]. For NF-κB activation, the phosphorylation of p65 is crucial for nuclear translocation and transcriptional activation [57]. Phosphorylation at different sites has specific significance; for example, phosphorylation at serine 536 is responsible for nuclear translocation, whereas phosphorylation at serine 529 is involved in transcriptional activity [57,108]. Conventional flow cytometry provides information regarding the degree of phosphorylation of the subunit, but not the localization of NF-κB [132]. One of the novel flow cytometry systems that has been used for measuring NF-κB signaling events is the ImageStream technology system (USA). This system provides high-resolution images of cells for up to 1000 cells per second. These images show the detailed cellular structure with the intracellular location of phosphorylation subunits of NF-κB using high-scale quantitative analysis [108]. Cellular images for the phosphorylation events were taken from HL-60 human promyelocytic leukemia cells.

### 2.9. Sample Consideration for NF-κB Detection

Albert Claude, who discovered subcellular fractionation in 1946, wrote, “The physiology of the cell cannot be fully understood unless we succeed in determining the constitution of its parts.” [133]. Since then, there have been many protocols developed for the extraction of different cellular components, such as whole nuclear proteins (nucleoplasmic proteins, nucleolar proteins, and histone proteins), and cytosolic proteins [75]. Apart from the nucleus, mitochondria and cytoplasm, a significant amount of NF-κB is present in dendrites and synaptic terminals as demonstrated by EMSA and Western blotting analysis for subcellular synaptosomal brain fractions [36,134]. Purified synaptosome were obtained by Percoll gradient. Reports of NF-κB in CNS neurons provide evidence that endogenous synaptic transmission activates this transcription factor. Several transcription factors are found in dendrites and axons. In dendrites, NF-κB, CREB, Stat3 and ELK-1 have been observed along with p65 in post-synaptic densities. Glutamate, NMDA and kainate stimulate NF-κB activation in the cerebellar granular neurons and this triggers the transport of the protein to the nucleus. This phenomenon has been observed in in vitro studies. Basal synaptic activity triggers NF-κB in cultured neurons from the cerebellum and hippocampus [135]. These different fractions serving as brain samples provide specific information on protein localization, phosphorylation events during transcriptional activation, and signaling for NF-κB [129]. These cellular protein extraction procedures should be optimized for cultured cells and tissue, such as for the number of cells and samples [75]. It is important to homogenize the cells to permit proper separation of the nucleus and the cytoplasm [75]. The most common method is separation by density gradient centrifugation with the use of detergents to solubilize membrane proteins [75].

Lysis permeabilizes the cell membranes and breaks down different cellular compartments. The lysis strength can be increased by increasing the detergent strength of the lysis buffer, which could be an important consideration for minimizing contamination across subcellular components [136]. This technique exposes cells to non-physiological hyperosmotic conditions that may often lead to leakage between the nucleus and the cytoplasm, hence contamination between the fractions [133]. This leakage will result in false-positive or false-negative bands in Western blots and EMSA blots [75]. Therefore, the purification and stability of sub-cellular fractions is a crucial step for these biochemical methods. There are several techniques used for the separation of cell compartments, once the disruption method has been selected; for example, it must be disruptive enough to break the cells and not damage the cell constituents [137]. Homogenization is a process that involves the physical disruption of cell membranes using shear force [138]. Homogenization is carried out by osmotic shock, mechanical force, sonication, or a combinations of these techniques [137]. Brain tissues are usually homogenized as they are difficult to break open and so Dounce homogenizers were commonly used [137]. Nowadays, however, there are a wide range of commercial kits available that are more convenient, such as NE-PER™ (nuclear and cytoplasmic extraction reagents from ThermoFisher Scientific, cell fractionation kit-standard from Abcam etc.), cell mitochondria isolation kit (8268, ScienCell Research Laboratory, Carlsbad, CA, USA) [100].

Furthermore, a short-cell fractionation method could be considered for studying NF-κB in neuronal or glial cultures [104,133]. This procedure avoids extended exposure time of cells to non-physiological conditions, minimizing the leakage between the nucleus and the cytoplasm and, thereby, increasing the purity and specificity. NF-κB is present in all cell types, but in low quantity in nuclear compartments in most cells, except in a few cell lines and tissues.

In the CNS, NF-κB is present in both neuronal and non-neuronal cells (e.g., astrocytes), where the most abundant forms are as mentioned. They are also present in synapses, which regulate NF-κB subcellular distribution and transcription [139]. In certain cell types where a low level of NF-κB is present, cell death pathways or growth arrest are triggered [140]. Phosphorylation events for NF-κB subunits are crucial for the function of the protein, as site-specific phosphorylation determines the stability, degradation, and binding of the protein to various factors. The phosphorylation of NF-κB subunits is an orchestrated and dynamic event, occurring within minutes of NF-κB activation [37]. This factor regulates gene-specific transcriptional activity that may downregulate or upregulate transcription of the specific gene [109]. The phosphorylation of the NF-κB p65 subunit is crucial for canonical pathway activity [141]. In the N-terminal Rel homology domain (Ser-205, Thr-254, Ser-276, Ser-281, and Ser-311) and the C-terminal transactivation domain (Thr-435, Ser-468, Thr-505, Ser-529, Ser-535, Ser-536, Ser-547) of p65, various phosphorylation sites have been identified [141].

Once activated, the protein undergoes post-translational modifications (PTMs), such as phosphorylation, acetylation, sumoylation, methylation, ubiquitination and nitrosylation [14,140]. These regulatory modifications can vary depending on the NF-κB-inducing stimulus [14]. These phosphorylation events determine the extent of NF-κB activity, such as RelA dephosphorylation by protein phosphatase 2A (PP2A), decreasing NF-κB activity [14,140]. Among all the subunits of NF-κB, p65 has received the most attention in terms of phosphorylation at two specific sites, S276 and S536. These phosphorylation events are mediated by several protein kinases, such as protein kinase A (PKA) with phosphorylation at site S276 of p65 [109]. The PKA catalytic subunit, PKA-C is bound to cytosolic IκBα in an inactive state in resting cells. PKA-C phosphorylates S276 at p65 when the protein kinase is activated upon the activation of the IKK complex and the degradation of IκBα. This phosphorylation interaction is promoted by A-kinase-interacting protein 1 (AKIP1) and leads to conformational change in p65, which then associates with CBP/p300, thus increasing p65 transcriptional activity [109]. Another phosphorylation event of PKA-NF-κB is through the interaction of NF-κB and glucocorticoid receptor (GR) pathways. This process requires PKA phosphorylation of S276 for repression of GR activity by p65 [109]. S276 phosphorylation of p65 stimulates the expression of a portion of NF-κB-regulated genes via the accumulation of cyclin-dependent kinase 9/cyclin T1 complexes to target promoters. On the other hand, the p50-HDAC-1 complex binds to DNA and inhibits NF-κB dependent gene expression in unstimulated cells. Upon stimulation of cells, the phosphorylated p65 enters the nucleus displacing the DNA-bound p50:HDAC-1 complex, promoting both acetylation and deacetylation for NF-κB-dependent transcription [142]. Recently, post-translational modification, such as lysine acetylation and deacetylation, have received much attention and have become one of the important domains for the regulation of nuclear NF-κB activity. The reversible process of acetylation is conducted by the following two enzymes: histone acetyltransferases (HAT) and histone deacetylases. About seven of the following lysines have been identified within RelA that are acetylated by p300/CBP or PCAF: lysines (K) 122, 123, 218, 221, 310, 314 and 315. This acetylation relates to different functions of NF-κB. The acetylation of K314 and K315 by p300 regulates specific sets of NF-κB target genes, whereas K310 is important for the full transcriptional potential of NF-κB, and is also required for the stability of RelA by preventing methylation at K314/315 and sustained NF-κB activity in cancer cells [69,143]. Along with these phosphorylation events, NF-κB translocation dynamics are measured for cell types [144]. One example is that the translocation time for macrophage cells was found to be within 40 min when activated with Toll-like receptors (TRL) ligands [144]. This shows nuclear residence and occupancy ability for macrophages, which was quite different than observed in fibroblasts [144]. In one study, the translocation of NF-κB was measured through EMSA and Western blotting when activated by TNF-α stimulation in fibroblasts [145]. As the IκBα levels decreased, NF-κB was seen within 5 min and then at 30 min when the cytoplasmic IκBβ and IκBЄ disappeared [145].

Given the significance of the NF-κB signaling processes in the pathophysiology of human disease, this molecule is regarded as a potent therapeutic target. More than 700 inhibitors of this pathway have been studied over the years [44]. Therefore, the use of inhibitors, such as phosphatase inhibitors, protein kinase inhibitors, protease inhibitors, and IκB ubiquitination blockers etc., is almost mandatory to study the NF-κB signaling pathway. Inhibitors can also be used at different stages of the NF-κB signaling process—upstream and downstream—for a more specific blockade of NF-κB activity.

## 3. Conclusions

NF-κB is one of the most explored proteins in scientific research; hence, an extraordinary amount of work has been put into understanding its activation process in different tissue and cell types. The biochemical detection methods and high-content analytical techniques have been important tools for investigating the underlying molecular mechanisms in signal transduction, from the stimulus to the activation of NF-κB, and for translocation, and post-translational modifications. Several other methods are now emerging for more accurate and direct measurements. These methods are available for the detection and interpretation of the protein’s biological actions in disease states, which can also generate unexpected challenges. Optimization is required in different sample types that vary among in vitro and in vivo models. However, in spite of the fact we are acquiring more insight about the function of this protein, more challenges are still surfacing. In this review, we have tried to survey some of the major challenges that have been documented by researchers and that we have uncovered in our lab over the years. Future studies will undoubtedly reveal new obstacles to overcome.

## Figures and Tables

**Figure 1 cells-10-01335-f001:**
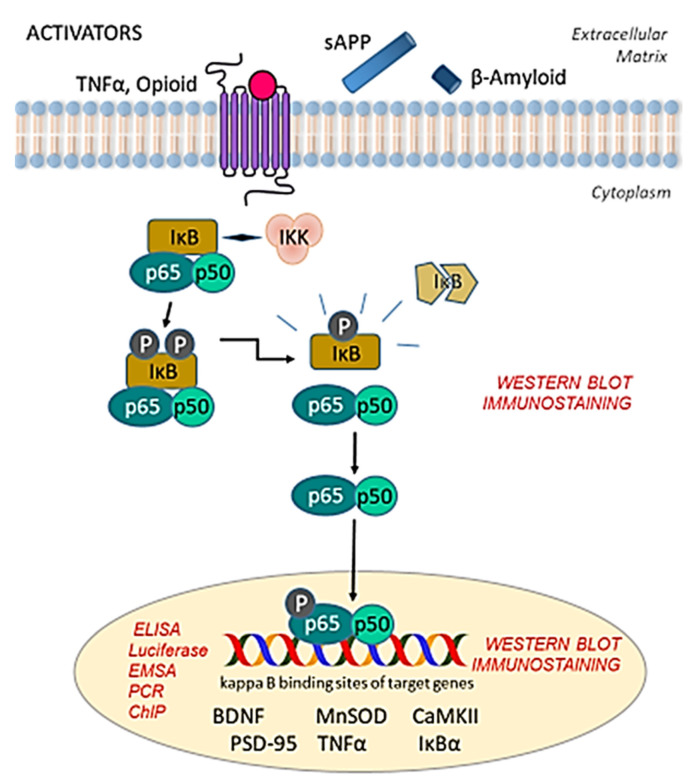
NF-κB signaling pathway. The transcription factor NF-κB is constitutively expressed in the central nervous system (CNS), where it can be activated by several stimuli, including TNFα, opioids, β-amyloid, and sAPP, to name a few. The NF-κB complex is sequestered in a dimer form (with p65/p50 dimers as the most common composition) in the cytoplasm, where it is bound by IκB. Upon activation by various stimuli, IKK interacts with the inhibitory IκB, resulting in phosphorylation, ubiquitination, and degradation of IκB, rendering the dimer free to translocate to the nucleus. Here, the dimer binds to kappa B binding sites of several gene targets that may be involved in cell survival in neurons or inflammatory pathways in glia. Dimers consisting of p65/p50 tend to be transcriptionally active, whereas p50 homodimers tend to suppress gene transcription. Red text indicates methods appropriate to investigate specific components of NF-κB signaling. BDNF = brain-derived neurotrophic factor; CaMKII = calcium-calmodulin kinase II; ChIP = chromatin immunoprecipitation; ELISA = enzyme-linked immunosorbent assay; EMSA = gel electrophoresis mobility shift assay; IκB = inhibitory κB protein; IKK = inhibitory κB protein kinase; MnSOD = manganese superoxide dismutase; PCR = polymerase chain reaction; PSD-95 = postsynaptic density protein-95; sAPP = secreted amyloid precursor protein; TNFα = tumor necrosis factor alpha.

**Figure 2 cells-10-01335-f002:**
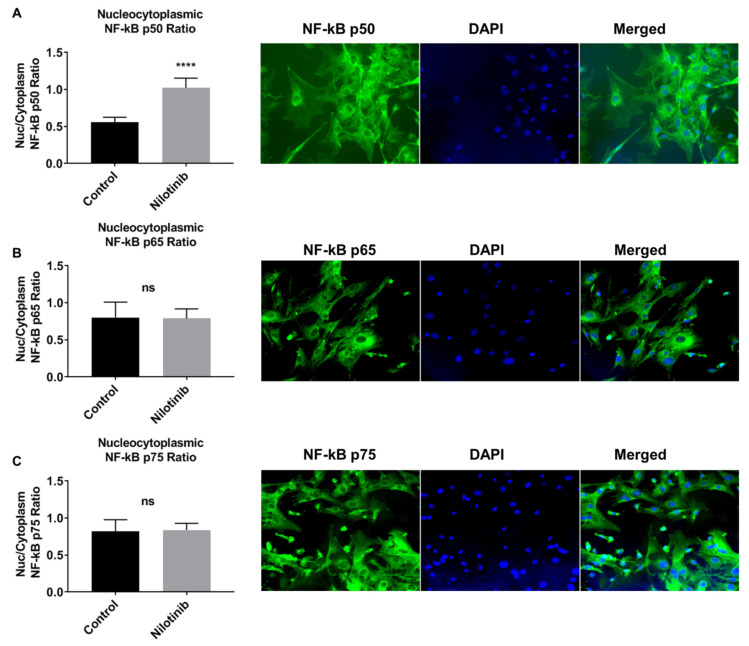
Nilotinib translocated the NF-κB p50 subunit into the nucleus of 3xTg-AD astroglia. Quantitative immunofluorescent nuclear/cytoplasmic ratios of NF-κB subunits (p50 (**A**), p65 (**B**), and p75 (**C**)) in 3xTg-AD astroglia. DAPI (blue) marks the nucleus. Images were captured at 100× magnification. Volume density of NF-κB subunits immunofluorescence was quantified using ImageJ software (**** *p* ≤ 0.0001); *n* = 5 per group analyzed by unpaired Student’s *t*-test. Reproduced from [104].

**Figure 3 cells-10-01335-f003:**
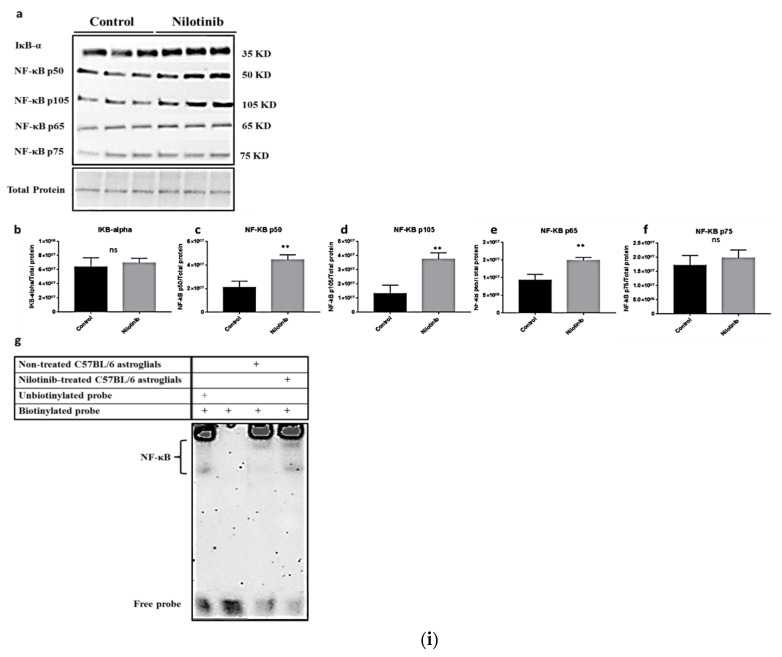
(**i**) Nilotinib significantly increased the activation of NF-κB and expression of NF-κB p50/p105 subunits in C57BL/6-WT astroglia. (**a**) Western blot experiments demonstrating relative levels of NF-κB subunits (p50, p105, p65, and p75), and IκB-α in cultured cortical astroglia derived from C57BL/6 in the presence and absence of 24 h 100 nM nilotinib treatment. (**b**–**f**) Relative quantification for protein levels of NF-κB subunits (p50, p105, p65, and p75) and IκB-α normalized to total protein. (**g**) Nuclear extract derived from nilotinib-treated and non-treated C57BL/6-WT astroglia were assayed for NF-κB activation by EMSA using a biotin-labeled oligonucleotide encompassing the NF-κB consensus motif. Results are expressed as mean ± SD of *n* = 6 per group (** *p* ≤ 0.01) analyzed by unpaired Student’s *t*-test. Reproduced from [104]. (**ii**) Nilotinib significantly increased the expression of NF-κB p50/p105 subunits and activation of NF-κB in 3xTg-AD astroglia. (**a**) Western blot experiments demonstrating relative levels of NF-κB subunits (p50, p105, p65, and p75), and IκB-α in cultured cortical astroglia derived from 3xTg in the presence and absence of 24 h 100 nM nilotinib treatment. (**b**–**f**) Relative quantification for protein levels of NF-κB subunits (p50, p105, p65, and p75) and IκB-α normalized to total protein. (**g**) Nuclear extract derived from nilotinib-treated and non-treated 3xTg-AD astroglia were assayed for NF-κB activation by EMSA using a biotin-labeled oligonucleotide encompassing the NF-κB consensus motif. Results are expressed as mean ± SD of *n* = 6 per group (* *p* ≤ 0.05) analyzed by unpaired Student’s *t*-test. Reproduced from [104].

**Table 1 cells-10-01335-t001:** Summary of the advantages and disadvantages of the methods used for NF-κB detection.

Method	Advantages	Disadvantages
Enzyme-Linked Immunosorbent Assay (ELISA)	High specificity and sensitivity because of antigen-antibody reaction High efficiency	High possibility of false-negative and false-positive results Expensive to prepare antibody Labor intensive Antibody stability
Luciferase Reporter Assay	Highly sensitive quantification Widely used for cell-based gene expression assays Large dynamic range of bioluminescence affords	Time consuming
Quantitative Real-Time Polymerase Chain Reaction	Rapid and sensitive Applicable for primary cells and frozen samples	High possibility of false-negative or false-positive results
Western Blotting	High sensitivity and specificity due to antigen-antibody reaction Has the ability to detect picogram level of proteins in a sample	High false or subjective results High cost High technical demand Requirement of a specific antibody to detect the protein of interest
Gel Electrophoresis Mobility Shift Assay (EMSA)	Simple, sensitive, and robust	Does not provide information regarding the location of nucleic acid sequences, molecular weight and identities of the protein in a protein-nucleic acid complex
Immunohistochemistry (Immunostaining)	High specificity due to antigen-antibody reaction High resolution Good signal amplification	Reaction bias Interruption bias Possibility of having a high background Species cross-reactivity
Chromatin Immunoprecipitation (ChIP)	Rapid and effective Determines interactions between DNA binding proteins, target genes and unknown DNA sequences	Not a quantitative approach Variability in crosslinking between DNA and target protein among samples Variability in crosslinking with antibody immunoprecipitation

## Data Availability

Permission was obtained to use the figure, Aging and Disease (Doi:10.14336/AD.2020.0910) for Figure 2 and Figure 3. Figure 1 and Table 1 are original data which is generated for this manuscript.

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
