# Peer review of "Challenges with Methods for Detecting and Studying the Transcription Factor Nuclear Factor Kappa B (NF-κB) in the Central Nervous System"

_cells, 2021, doi:10.3390/cells10061335_

Round 1

Reviewer 1 Report

This is an interesting and comprehensive review of techniques to study NF-kB, particularly in the nervous system. The introduction about the importance of this pathway of transcription regulation in the brain is adequate and the descriptions of the techniques are ample.

However, I have some suggestions that I hope will improve the manuscript.

1) Sentence from line 56 to 61. To my knowledge, CREB is not a gene target of NF-kB as the sentence seems to affirm. If so, a paper citation showing this should be offered. The authors cite a review in which the relationship between CREB and NF-kB is described, but no information about CREB gene regulation by NF-kB is indicated.

2) Between the different techniques described, I suggest to the authors to discuss about the use of chromatin immunoprecipitation (ChIP) assay as a powerful method to precisely determine the presence of NF-kB in the promoter of the target genes, such as Federman et al., 2013.

3) It should be interesting to describe the use of subcellular fractionations techniques other than nuclear and cytosolic extracts, such as synaptosome isolation and fractionation in synaptosomal content and membrane content, such as Salles et al., 2015.

4) Regarding post-translational modifications of NF-kB, p65 phosphorilation by PKA is a very important modification in the activation of the transcription factor, as well in the association to the histone acetylase CBP, and I believe deserves a discussion. Similarly, the modification of p65 by acetylation in lysisne 221, 310, etc. is an important regulation of the transcription factor function.

Author Response

Thank you for sharing your suggestions on improving the manuscript. I have made the required changes and included new details in the manuscript according to your suggestion

1) Sentence from line 56 to 61 (now lines 61 to 65). To my knowledge, CREB is not a gene target of NF-kB as the sentence seems to affirm. If so, a paper citation showing this should be offered. The authors cite a review in which the relationship between CREB and NF-kB is described, but no information about CREB gene regulation by NF-kB is indicated.

- Thank you for the clarification. CREB is indeed not considered a direct gene target of NF‑κB but is an important transcriptional regulator of learning and memory that is responsive to NF-κB activity. This section has been edited to reflect this (page 2, line 61-65).

2) Between the different techniques described, I suggest to the authors to discuss about the use of chromatin immunoprecipitation (ChIP) assay as a powerful method to precisely determine the presence of NF-kB in the promoter of the target genes, such as Federman et al., 2013.

-The ‘Chromatin immunoprecipitation (ChIP)’ assay has been included in the manuscript along with other techniques. This new technique description can be found in the manuscript on page 14, line 580 to 626. This section includes what the technique is and how it has proven to be useful in accurate detection of  NF‑κB in comparison to other available techniques. The advantages, disadvantages and limitations of the assay have also been discussed. Furthermore, different types of samples are mentioned for which this assay has been used.

3) It should be interesting to describe the use of subcellular fractionations techniques other than nuclear and cytosolic extracts, such as synaptosome isolation and fractionation in synaptosomal content and membrane content, such as Salles et al., 2015.

-Along with the description of nuclear and cytosolic extract description under the topic, ‘Sample Consideration for NF-κB Detection’ on page 18 line 685 to 695, brief description on synaptosome has been included. This section reflects on the presence of NF-κB in this fraction and how the activation process of the transcription factor can be detected.

4) Regarding post-translational modifications of NF- κB, p65 phosphorylation by PKA is a very important modification in the activation of the transcription factor, as well in the association to the histone acetylase CBP, and I believe deserves a discussion. Similarly, the modification of p65 by acetylation in lysisne 221, 310, etc. is an important regulation of the transcription factor function.

-In the manuscript a detailed discussion on p65 phosphorylation by PKA, histone acetylase CBP and modification of p65 by acetylation in the Lysine 221, 210 etc, has been added on page 19, line748-772, in the manuscript.

Reviewer 2 Report

The review attempts to summarize and discuss the methods that are available to study the NF-kappa B pathway in the CNS. After a short introduction in this pathway, it covers the relevant techniques and gives examples for use with CNS cells or tissue samples. Even though the review is already quite long, the review remains a bit superficial and the information about NF-kappaB in the CNS and the detection methods is widely distributed and thereby “hidden” in the text.

The manuscript would profit from a figure displaying the different components of the NF-kappa B pathway and which detection method can be applied for which component. A table summarizing key findings of NF-kappaB pathway specifics in CNS cells would make this information more accessible.

Also, figures showing examples of results that can be obtained e.g. with EMSA, Western Blot or immunostaining are missing. Especially a figure showing p65 nuclear translocation (e.g. after TNF treatment) in neurons, astrocytes and microglia by immunofluorescence would greatly improve the manuscript. Also, a figure, sketch or graph indicating how translocation can be quantified can complement the paper.

Furthermore, information about the time scale of reactions (minutes, hours, days) would be very helpful for readers. In the current version, this information is only given for phosphorylation events.

In the RT-qPCR section, a discussion of reference genes or housekeeper genes to be used in brain samples or brain cells is missing.

Abstract:

The last two sentences just describe what is summarized in the review. It would be more interesting for the reader to find short conclusions about the different methods to detect NF-kB and precise recommendations.

Introduction

p. 2 lines 70-71: Ionizing radiation can also activate NF-kappa via the genotoxic stress induced pathway which starts at DNA double strand beaks in the cell nucleus and involves ATM, NEMO, PIASy and other proteins transferring the signal from the nucleus to the cytoplasm. This could be of interest for radiotherapy of brain tumors and is therefore also relevant for this review.

p. 5 lines 217-219: “The luciferase reporter assay is a traditional method used for assessing signaling pathways to measure NF-κB levels in in-vitro models [81] [83]”

=> “The luciferase reporter assay is a traditional method used for assessing signaling pathways to measure levels of activated NF-κB in in-vitro models [81, 83]”

p. 5 lines 223-224: “Transcriptional activation of NF-κB can be detected by placing the luciferase gene under the control of promotor for NF-κB [86].” => It is unclear what “Transcriptional activation of NF-κB” means in this context: Activation of the transcription of a gene encoding NF-kappaB subunits? Or activation of NF-kappaB, resulting in its translocation in the nucleus and binding to promoters containing NF-kappaB binding sites? In this case, the luciferase gene would be under control of a promoter containing NF-kappaB binding sites.

p. 5 lines 225-226: “This assay has proven to be extremely useful in observing the effects of drugs on transcriptional levels [87].” => Reference [87] discusses effects of wear particles, not of drugs.

p. 5 line 237: “Immunostaining (a.k.a., immunofluorescence)” => “Immunostaining (immunofluorescence, immunohistochemistry and immunocytochemistry)”

Tables:

Table 1:

The layout of the table can be improved by removing the dots.

Also, mentioning what a specific method detects, e.g. NF-kappaB translocation or expression of NF-kappaB target genes, would be helpful. Suggestions which (target) genes or proteins or protein modifications indicate NF-kappaB activation would further improve the information given to the reader.

“Luciferase Reporter Assay - Detailed visualization method for specific characterization the distribution and quantity of a specific protein and/or molecule in tissues and cultured cells.” => The description seems to apply to immunostaining.

“pictogram” => “picogram”

Minor points:

In some sentences, there seem to be too many blanks. Please check.

p. 2 line 56: [20] [21] => [20, 21]

p. 2 line 83: constituent => constitutive?

p. 3 line 127: [49] [50] => [49, 50]

p. 4 line 155: [55] [67] => [55, 67]

p. 4 line 156: [69] [68] => [69, 68]

p. 5 line 228: [88] [83] => [88, 83]

p. 5 line 235: [81, 89] [81] => [81, 89]

p. 5 line 245: “processing the protein of interest [92] [91]” => “detecting the protein of interest [92, 91]”

p. 6 lines 289-291: “The non-canonical pathway is activated by tumor necrosis factor (TNF) receptor superfamily members and canonical pathway activation of NF-κB dimers [101]. These dimers have RelB and p52 subunits that then cause IKKα- and NIK-dependent proteasomal processing of p100 to p52 [100].“ => The sequence of events seems to be reversed in this sentence, and the role of canonical NF-kappaB dimers in the non-canonical pathway is surprising. Please check.

p. 6 lines 308-309: “A study was conducted targeting p50 and p65 and were tested to determine their specificity towards their antigen.” => A word seems to be missing in this sentence, maybe “antibody”?

p. 6 lines 323-324: “qPCR is used for the detection and quantification of gene expression.” => “RT-qPCR is used for the detection and quantification of gene expression.”

p. 10 lines 429: “exposition time” => “exposure time”

p. 10 lines 435-436: “They are also present in synapses, which regulates …” => “They are also present in synapses, which regulate …”

p. 10 lines 437-438: “cell death pathways or growth arrest is triggered.” => “cell death pathways or growth arrest are triggered.”

p. 10 lines 448-449: “Once activated the protein undergoes post-translational modifications (PTMs), such as phosphorylation, acetylation, and methylation [121].” => Sentence seems to be truncated. Please check.

p. 11 line 452: “NF-kB” => Greek symbol for kappa instead of “k”

References:

p. 11 line 486: “NF-kB” => Greek symbol for kappa instead of “k”

p. 11 line 489: “Zn2+” => “2+” superscript

p. 11 line 499: “NFkB” => Greek symbol for kappa instead of “k”

p. 12 line 525: “NF-kB” => Greek symbol for kappa instead of “k”

p. 14 line 661: “NF-kB” => Greek symbol for kappa instead of “k”

Author Response

Thank you for sharing your suggestions on improving the manuscript. I have made the required changes and included new details in the manuscript according to your suggestion.

The manuscript would profit from a figure displaying the different components of the NF-kappa B pathway and which detection method can be applied for which component. A table summarizing key findings of NF-kappaB pathway specifics in CNS cells would make this information more accessible.

-A schematic diagram has been included in the manuscript detailing the NF-κB signalling pathway and its components. The different detection techniques (mentioned in the manuscript) are also included in the diagram and correspond to the pathway components. This schematic figure gives a clear overview of the manuscript which, we feel, eliminates the need for the summarized table (as suggested in point 2 below). This figure (Figure 1) is on page 5 of the manuscript.

Also, figures showing examples of results that can be obtained e.g. with EMSA, Western Blot or immunostaining are missing. Especially a figure showing p65 nuclear translocation (e.g. after TNF treatment) in neurons, astrocytes and microglia by immunofluorescence would greatly improve the manuscript. Also, a figure, sketch or graph indicating how translocation can be quantified can complement the paper.

-We have included experimental data showing NF-κB nuclear translocation detected via immunostaining (pg. 9) as well as the expression of NF-κB p50/p105 subunits in C57BL/6-WT vs 3xTg-AD astroglia, detected through western blotting (please see additional figures on pages 11-12). These figures were previously published from an open access journal and have been referred to as such. We look forward to working with the copyright officer to confirm that the statements regarding these previously published works meet the journal’s guidelines upon final acceptance (e.g., copyright statements, etc).

Furthermore, information about the time scale of reactions (minutes, hours, days) would be very helpful for readers. In the current version, this information is only given for phosphorylation events.

-The scale of reactions has been mentioned in the manuscript and can be found on page 19 lines 772-780.

In the RT-qPCR section, a discussion of reference genes or housekeeper genes to be used in brain samples or brain cells is missing.

  • A detailed discussion on reference genes or housekeeping genes that are used for brain samples have been included in the manuscript on page 13-14, line 525-559.

    Introduction

    1. 2 lines 70-71 (now p. 3, lines 103-134): Ionizing radiation can also activate NF-kappa via the genotoxic stress induced pathway which starts at DNA double strand beaks in the cell nucleus and involves ATM, NEMO, PIASy and other proteins transferring the signal from the nucleus to the cytoplasm. This could be of interest for radiotherapy of brain tumors and is therefore also relevant for this review.

    -Details regarding the DNA damage induced NF-κB activation via ATM, NEMO, PIASy and other proteins have been described in the manuscript. Also, different stressors are involved in activation of NF-κB activation such as ionizing radiation and DNA damage drugs are included. This section also describes how this activation can be therapeutic in brain tumours such as Glioblastoma.

    The following has been corrected in the manuscript-

    • 5 lines 217-219 (now p.7 lines 316-318) : “The luciferase reporter assay is a traditional method used for assessing signaling pathways to measure NF-κB levels in in-vitro models [86] [88]”

      => “The luciferase reporter assay is a traditional method used for assessing signaling pathways to measure levels of activated NF-κB in in-vitro models [86, 88]”

    • 5 lines 223-224: “Transcriptional activation of NF-κB can be detected by placing the luciferase gene under the control of promotor for NF-κB [86].” => It is unclear what “Transcriptional activation of NF-κB” means in this context: Activation of the transcription of a gene encoding NF-kappaB subunits? Or activation of NF-kappaB, resulting in its translocation in the nucleus and binding to promoters containing NF-kappaB binding sites? In this case, the luciferase gene would be under control of a promoter containing NF-kappaB binding sites.

    - Thank you for the clarification. To prevent confusion this sentence has been removed.

    • 5 lines 223-224: (now p.7 lines 323-324): “This assay has proven to be extremely useful in observing the effects of drugs on transcriptional levels [87].” => Reference [87] discusses effects of wear particles, not of drugs.

    • 5 line 237 (now p. 7 lines 334) p. 7: “Immunostaining (a.k.a., immunofluorescence)” => “Immunostaining (immunofluorescence, immunohistochemistry and immunocytochemistry)”

    Tables

    As suggested, the dots of the table have been removed to improve the layout of the table.

    The description for Luciferase assay has been changed to make it more specific to the assay itself and to avoid confusion with immunostaining.

    Regarding the inclusion of what a specific method detects in the table, this information has been described clearly in the text under each technique description. This table has been provided to give an overview of the advantages and disadvantages for each assay, in the form of a simple comparison.

    Minor mistakes such as ‘‘pictogram-picogram’’ has been corrected.

    Minor points

    In this section, I am referring to the updated numbers to address the comments. All suggested changes have been included.

    1. 2 line 56: [20] [21] => [20, 21]

      p. 2 line 83 (now line 88): constituent => constitutive?

      p. 3 line 127 (now line 133-134): [49] [50] => [49, 50] ,
    2. 4 line 165 (now p.4 line 165): [54] [55] => [54, 55], now [55, 56]
      p. 4 line 155 (now p.6 line 252): [55] [67] => [55, 67], now [61,73]
    3. 4 line 156 (now p.6 line 253-254): [69] [68] => [69, 68], now [74, 75]
    4. 5 line 228 (now p.7 line 318): [88] [83] => [88, 83], now [87,89])
    5. 5 line 235 (now p. 7 line 332): [81, 89] [81] => [81, 89], now [87, 96]

      p. 5 line 245 (now p.7 line 343 ): “processing the protein of interest [92] [91]” => “detecting the protein of interest [92, 91]”, now [98,99]):
    6. 6 lines 289-291 (now p.10 line 441-447): “The non-canonical pathway is activated by tumor necrosis factor (TNF) receptor superfamily members and canonical pathway activation of NF-κB dimers [101]. These dimers have RelB and p52 subunits that then cause IKKα- and NIK-dependent proteasomal processing of p100 to p52 [100].“ => The sequence of events seems to be reversed in this sentence, and the role of canonical NF-kappaB dimers in the non-canonical pathway is surprising. Please check.

    1. p. 6 lines 308-309 (now p.13 lines 506-507): “A study was conducted targeting p50 and p65 and were tested to determine their specificity towards their antigen.” => A word seems to be missing in this sentence, maybe “antibody”?,.

      6 lines 323-324 (now p.13 line 553.): “qPCR is used for the detection and quantification of gene expression.” => “RT-qPCR is used for the detection and quantification of gene expression.”

      p. 10 lines 429 (now p. 18 lines 723-724): “exposition time” => “exposure time”

      p. 10 lines 435-436 (now p. 18-19 lines 729-730): “They are also present in synapses, which regulates …” => “They are also present in synapses, which regulate …”

      p. 10 lines 437-438 (now p. 19 lines 742-744): “cell death pathways or growth arrest is triggered.” => “cell death pathways or growth arrest are triggered.”

      p. 10 lines 448-449 ( now p. 19 lines 699-702): “Once activated the protein undergoes post-translational modifications (PTMs), such as phosphorylation, acetylation, and methylation [14,135].” => Sentence seems to be truncated. Please check.

      p. 11 line 452 (now p. 19 line 788): “NF-kB” => Greek symbol for kappa instead of “k”

    References-

    p. 11 line 486 (now p.20 line 816): “NF-kB” => Greek symbol for kappa instead of “k”

    p. 11 line 489 (now p.20 line 821): “Zn2+” => “2+” superscript

    p. 11 line 499 (now p.21 line 831): “NFkB” => Greek symbol for kappa instead of “k”

    p. 12 line 525 (now p.21 line 864): “NF-kB” => Greek symbol for kappa instead of “k”

    Thank you for pointing out these typographical errors, correction of some information and spacing between references which have been corrected.

Round 2

Reviewer 2 Report

Please find the comments in the attached Word file

Author Response

Response to Reviewers

cells

Manuscript Number: cells-1162681-R1

Title: Challenges with Methods for Detecting and Studying the Transcription Factor Nuclear Factor Kappa B (NF-κB) in the Central Nervous System

Recommendation: Minor revision

Thank you for your suggestions on improving the manuscript. We have made all the required changes as suggested.

  • 9 Figure 2: The first line of the legend has moved above the figure and should be moved below. Please use the Greek symbol for kappa in NF-kB in the legend and in the figure.

  • The location of Figure 2 in the manuscript has been rearranged and the Greek symbol for NF-κB has been changed in the figure legend.

  • Figure 3: The legend is on the next page instead of below the figure. Please use the Greek symbol for kappa in NF-kB in the figure.

  • Figure 2 and Figure 3 has been merged into one figure as suggested by the reviewer.

  • 3 on p. 10 and Fig. 4 on p. 12 are very similar, showing wt and 3xTg astroglia – combining them to one figure with one legend would be less confusing.

- As suggested, Fig. 3 on p. 10 and Fig. 4 on p. 12 has been combined into one figure, Figure 3 (i) and (ii) on p. 11-12.

4)  The following errors have been fixed-

  1. 12 line 496 western blot analyses[112]. => western blot analyses [112].

  1. 12 lines 505-506 A study was conducted targeting p50 and p65 to determine their specificity towards antibodies. => A study was conducted to determine the specificity of antibodies towards p50 and p65.

  1. 13 lines 534-535 gglyceraldehyde-3-phosphate => glyceraldehyde-3-phosphate

  1. 14 lines 544 brains samples => brain samples

  1. 14 lines 571 NF-kB p100 protein => Please use the Greek symbol for kappa in - NF-kB. This was changed.

  1. 14 lines 596 Also the ant-apoptotic function => Also the anti-apoptotic function

  1. 14 lines 596 TNF-a = Please use the Greek symbol for alpha in - TNF-a. This was changed.

  1. 19 lines 615 inetractions => interactions. Also changed.

  1. 18 lines 684 synatic terminals => synaptic terminals. Changed.

  1. 18 lines 686 Reports of NF-kB in CNS neurons=> Please use the Greek symbol for kappa in NF-kB. Was changed.

  1. 19 lines 756-757 S276 phosphorylation of p65 stimulates expression of a portion of NF-kB-

regulated => S276 phosphorylation of p65 stimulates expression of a portion of NF-kB-regulated genes. Please use the Greek symbol for kappa in - NF-kB. This was changed.

  1. 19 lines 758-761 HDAC-1 bound to DNA S276A p65 => Do you mean: HDAC-1 bound to DNA on which S276-phosphoylated p65 binds? What does the “A” in S276A mean? We changed this to make this more clear and eliminated the “A” to eliminate confusion.

  1. 19 lines 756 NF-kB dependent transcription => NF-kB dependent transcription. Please use the Greek symbol for kappa in - NF-kB. This was changed.

  1. 19 lines 755-756 promoting acetylation, deacetylation for NF-kB dependent transcription => acetylation, deacetylation or both? This was changed to promoting both acetylation and deacetylation.

  1. 19 lines 764 nuclear NF-kB activity => Please use the Greek symbol for kappa in - NF-kB. This was changed.

  1. 20 lines 781-782 this molecule provides a potent therapeutic target for scientists => this molecule is regarded as a potent therapeutic target. Was changed.

  • Table 1 The description is still confusing.

“The description for Luciferase assay has been changed to make it more specific to the assay itself and to avoid confusion with immunostaining.”

  • A transient or stable transfection of cells necessary or transgenic animals expressing the luciferase reporter construct are required.
  • Luciferase can be expressed under the control of a promoter of interest to monitor activity of this promoter under different conditions.
  • Luciferase might be expressed also as a fusion protein with a protein of interest under control of a promoter of choice (constitutive active, inducible) to monitor the level of the protein of interest.

-Thank you so much for your comments. The Luciferase assay has now been described in the manuscript and the table is a representation of the advantages and disadvantages of the assay. This table is not a comparison between the different techniques.